# An Empirical Investigation to the "Skew" Phenomenon in Stock Index Markets: Evidence from the Nikkei 225 and Others

**Yizhou Bai** [1] and **Zhiyu Guo** [2,*]

1   School of Mathematical Sciences, Nankai University, Tianjin 300071, China; baiyizhou@mail.nankai.edu.cn
2   Business School, Nankai University, Tianjin 300071, China
*   Correspondence: zhiyuguo@mail.nankai.edu.cn

**Abstract:** The skew processes have recently received much attention, owing to their capacity to describe controlled dynamics. In this paper, we employ the skew geometric Brownian motion (SGBM) to depict nine major stock index markets. The skew process not only shows us where the "support" and "resistance" levels are, but also how strong the force is. However, the densities of the skew processes make it challenging to estimate the parameters in a convenient manner. For the sake of overcoming this challenge, we adopt a Bayesian approach, which plays an important role in allowing us to estimate the parameters by conditional probability densities without having to evaluate complex integrals. Furthermore, we also propose the likelihood ratio tests and significance tests for the skew probability. In the empirical study, our findings reveal that skew phenomenon exists in the global stock markets and that the SGBM model works better than the traditional GBM model, as well as performing competitively, compared to the GBM-jump model (GBM-J) and Markov regime switching GBM model (GBM-MRS). In addition, we explore the possible reasons behind the skew phenomenon in stock markets, the price clustering phenomenon and herd behaviors can help to explain the skew phenomenon.

**Keywords:** skew geometric Brownian motion; skew phenomenon; stock index

## 1. Introduction

For decades, a classical problem has been how to describe the dynamics of stock prices. The most basic model used to describe stock prices is geometric Brownian motion (GBM), which was first used by Black and Scholes [1]. However, there are many empirical features of market data that cannot be captured by GBM. Many researchers have proposed different approaches to modeling phenomenon in the market. For example, Merton [2] proposed a jump-diffusion process to capture the extraordinary magnitude changes of the asset price and Hamilton [3] introduced regime switching to capture structural changes in time series data. In this paper, we focus on another market phenomenon, which is called long-duration asset prices or price clustering. This phenomenon has been observed by some researchers. Sonnemans [4] found that the price of the stocks on the Dutch stock market tended to cluster at round numbers (e.g., 10, 20, 30, etc. or 5, 15, 25, etc.) and were affected by the round number price barriers (prices passed less frequently round numbers than other numbers). Similar phenomenon was also observed in China (Brown and Mitchell [5]; Hu et al. [6]) and Japan (Aşçioğlu et al. [7]). The phenomenon can be described by the skew diffusion process. Itô and McKean [8] first introduced the skew Brownian motion (SBM), a process that behaves similarly to the conventional Brownian motion before arriving at the skew level. Once it hits a skew level $a$, its excursion from $a$ has a probability of $p$ to go upward and a probability of $1 - p$ to go downward. Due to this property of skew process,

it is very suitable for describing the price clustering phenomenon. We notice that skew processes can also illustrate the strength of the phenomenon through the skew probability $p$. It is a more precise characterization of the phenomenon than price clustering. Thus, we name the phenomenon described by the skew process as skew phenomenon.

The mathematics of the SBM was explored, for instance, by Walsh [9], Le Gall [10], Barlow et al. [11], Ouknine [12], Lejay and Martinez [13], Ramirez et al. [14], Ramirez [15], and Appuhamillage and Sheldon [16]. The diverse applications of SBM in many fields have been investigated, such as in geophysics (Appuhamillage et al. [17]; Atar and Budhiraja [18]; Song et al. [19]), molecular biology (Bossy et al. [20]), population ecology (Cantrell and Cosner [21]; Min et al. [22]; Barahona et al. [23]), and so on. The SBM has also been applied to financial modeling by several scholars; for example, Corns and Satchell [24] and Gairat and Shcherbakov [25] used them to model the dynamics of the underlying asset and price derivatives. Although Rossello [26] argued that there is arbitrage in SBM models, Zhu and He [27] demonstrated that, by carefully choosing a risk-neutral measure, the SBM could be consistent with no-arbitrage property.

In the existing literature on SBM, much effort has been made in exploring the mathematical properties of SBM or applying them to model the underlying asset and price derivatives. However, empirical testing of SBM has received much less attention. Is there, indeed, the skew phenomenon present in the market? Can skew geometric Brownian motion (SGBM) outperform other models when applied to modeling the dynamics of asset price? We attempt to answer these questions in this paper. The contributions of this paper are four-fold. First, we apply a Bayesian estimation approach to estimate the parameters. To do this, the local time in the model is removed by a piecewise transform. Second, we test whether skew phenomenon is present in the global stock markets by checking whether the skew probability $p$ is significantly different from 0.5. Third, we compare the performance of SGBM with three other models, the basic GBM model, GBM-jump model (GBM-J) and Markov regime switching GBM model (GBM-MRS). Fourth, we show that the skew phenomenon and the price clustering phenomenon can confirm with each other; the skew phenomenon is also related to the herd behavior. In this point, we try to explain why the skew phenomenon exists.

The remaining part of this paper is organized as follows. Section 2 presents the SGBM model and its piecewise transform. Section 3 specifies the methods for evaluating the model's performance. Section 4 describes the data and empirical results. In Section 5, we try to find some theories to explain the existence of skew phenomenon. Finally, the conclusion is presented in Section 6.

## 2. The Model

Let $S_t$ be the stock price at time $t$. Assume that $S_t$ follows the SGBM

$$\frac{dS_t}{S_t} = \widetilde{\mu}dt + \sigma dW_t + (2p-1)\,d\hat{L}_t^S\,(a)\,, \tag{1}$$

where $\hat{L}_t^S\,(a)$ is the symmetric local time of the process $S_t$ at the "skew level" $a$, and the skew probability $p$ is the probability of moving upward when $S_t$ hits the level $a$. If we set $\mu = \widetilde{\mu} - \frac{1}{2}\sigma^2$ and $X_t = \ln S_t$, then we get that $X_t$ follows

$$dX_t = \mu dt + \sigma dW_t + (2p-1)\,d\hat{L}_t^X\,(\ln a)\,. \tag{2}$$

It is not difficult to obtain that $\hat{L}_t^X\,(\ln a)$ is the symmetric local time of the process $X_t$ and the "skew level" of $X_t$ is $\ln a$. To remove the local time, similar to Harrison and Shepp [28], we define a function $G(x)$ by

$$G\,(x) = \begin{cases} (1-p)\,(x-\ln a) + \ln a, & \text{if } x \geq \ln a, \\ p\,(x-\ln a) + \ln a, & \text{if } x < \ln a. \end{cases} \tag{3}$$

Then, we obtain its inverse function $H(x)$ as

$$
H(x) = \begin{cases} \dfrac{x}{1-p} - \dfrac{p \ln a}{1-p}, & \text{if } x \geq \ln a, \\ \dfrac{x}{p} - \dfrac{(1-p) \ln a}{p}, & \text{if } x < \ln a. \end{cases}
\tag{4}
$$

Consider $Y_t = G(X_t)$. Then,

$$
Y_t = \begin{cases} (1-p)(X_t - \ln a) + \ln a, & \text{if } X_t \geq \ln a, \\ p(X_t - \ln a) + \ln a, & \text{if } X_t < \ln a. \end{cases}
\tag{5}
$$

The inverse transform $X_t = H(Y_t)$ can be expressed as

$$
X_t = \begin{cases} \dfrac{Y_t}{1-p} - \dfrac{p \ln a}{1-p}, & \text{if } Y_t \geq \ln a, \\ \dfrac{Y_t}{p} - \dfrac{(1-p) \ln a}{p}, & \text{if } Y_t < \ln a. \end{cases}
\tag{6}
$$

As the function $G(x)$ is the difference of two convex functions, we apply the generalized Itô formula to $Y_t = G(X_t)$ to obtain:

$$
dY_t = \begin{cases} (1-p)(\mu dt + \sigma dW_t), & \text{if } Y_t \geq \ln a, \\ p(\mu dt + \sigma dW_t), & \text{if } Y_t < \ln a. \end{cases}
\tag{7}
$$

The discretized version of Equation (7) can be expressed as:

$$
\Delta Y_{t_i} = \begin{cases} (1-p)\left(\mu \Delta t + \sigma \sqrt{\Delta t}\epsilon_{t_i}\right), & \text{if } Y_{t_i} \geq \ln a, \\ p\left(\mu \Delta t + \sigma \sqrt{\Delta t}\epsilon_{t_i}\right), & \text{if } Y_{t_i} < \ln a, \end{cases}
\tag{8}
$$

where $\Delta Y_{t_i} = Y_{t_{i+1}} - Y_{t_i}$ and $\{\epsilon_{t_i}\}_{i=1}^N$ are independent and standard normally distributed. Denote $N_1$ (respectively, $N_2$) as the set in which the sample value is bigger (respectively, smaller) than the skew level $\ln a$ (i.e., $N_1 := \{i : i = 1, \cdots, N, x_{t_i} \geq \ln a\}$ and $N_2 := \{i : i = 1, \cdots, N, x_{t_i} < \ln a\}$). Let $n_1, n_2$ be the number of $i$ included in the set $N_1$ and $N_2$, respectively. Obviously, $n_1 + n_2 = N$. Define $y_{t_i} := \Delta Y_{t_i}$. Then, the likelihood function of Equation (8) can be written as:

$$
\begin{aligned}
l(X|\Theta) &= \left(\frac{1}{(1-p)\sqrt{2\pi\sigma^2 \Delta t}}\right)^{n_1} \exp\left\{-\frac{1}{2(1-p)^2 \sigma^2 \Delta t} \times \sum_{i \in N_1} [y_{t_i} - (1-p)\mu \Delta t]^2\right\} \\
&\quad \times \left(\frac{1}{p\sqrt{2\pi\sigma^2 \Delta t}}\right)^{n_2} \exp\left[-\frac{1}{2p^2 \sigma^2 \Delta t} \times \sum_{i \in N_2} (y_{t_i} - p\mu \Delta t)^2\right],
\end{aligned}
\tag{9}
$$

where $\Theta$ is the set of all parameters $\{\mu, \sigma^2, a, p\}$ in Equation (8) and $X$ represents the sample data.

## 3. Model Performance Evaluation

### 3.1. Significance Test of Skew Probability p

As SBM reduces to standard Brownian motion when $p = 0.5$, the skew model does not make any sense if the estimated $\hat{p}$ is not significantly different from 0.5. Thus, we need to test the significance of the estimated skew probability $\hat{p}$. The null hypothesis $H_0$ is $\hat{p} = 0.5$, and the alternative hypothesis $H_1$ is $\hat{p} \neq 0.5$. The other parameters are assumed to be the same under the two hypotheses. Under the null hypothesis, the likelihood function is

$$l\left(X|\Theta_0\right) = \left(\frac{2}{\sqrt{2\pi\sigma^2\Delta t}}\right)^N \exp\left[-\frac{2}{\sigma^2\Delta t}\times\sum_{i\in N}\left(y_{t_i}-\frac{1}{2}\mu\Delta t\right)^2\right]. \tag{10}$$

The likelihood ratio is defined as follows:

$$\lambda(X) = 2(\sup\ln l(X|\Theta_1) - \sup\ln l(X|\Theta_0)), \tag{11}$$

where $X$ is the sample data, $l(X|\Theta_1)$ is the likelihood function under $H_1$, and $l(X|\Theta_0)$ is the likelihood function under $H_0$. When $\lambda$ is large enough, we reject $H_0$ and accept $H_1$; then, we can consider the alternative model to be better than the null model.

We need to know the distribution of $\lambda$ under $H_0$ in order to decide whether to accept the null hypothesis or reject it at a specified significance level; however, it is difficult to obtain an analytical solution of the distribution. Thus, we apply the bootstrap method to obtain an approximation of the distribution. First, we get the parameters of standard BM model by maximum likelihood estimation. Then, a simulated sample $X^*$ can be obtained using the estimated model and, thus, we can calculate a value of $\lambda$ based on $X^*$. Repeat this process 10,000 times, we can get 10,000 $\lambda(X^*)$. Finally, we can get the *p*-value of $\lambda(X)$, which is the probability that $\lambda(X^*)$ is bigger than $\lambda(X)$.

### 3.2. Comparison of Model Performance

After verifying previous procedure, in this part, we compare the SGBM defined by Equation (1) with the commonly-used models, to see which model fits the best. We focus on the following three models.

- GBM:

$$\frac{dS_t}{S_t} = \widetilde{\mu}dt + \sigma dW_t, \tag{12}$$

- GBM-J:

$$\frac{dS_t}{S_t} = \widetilde{\mu}dt + \sigma dW_t + d(\sum_{i=1}^{N(t)} Z_i), \tag{13}$$

where $Z \sim \mathcal{N}(\mu_J,\sigma_J^2)$, $N(t)$ is a Poisson process with intensity $\lambda$, and $W_t$, $Z_v$ and $N(t)$ are independent of each other.

- GBM-MRS:

$$\frac{dS_t}{S_t} = \widetilde{\mu}_{s_t}dt + \sigma_{s_t}dW_t, \tag{14}$$

where $s_t$ is a two state continuous-time Markov process which is independent of $W_t$.

The null hypothesis is that the GBM (respectively, GBM-J and GBM-MRS) is suitable for fitting the data, and the alternative hypothesis is that the SGBM fits the data better. The test statistic $\lambda(X)$ defined by Equation (11) is also adopted. We obtain the *p*-value from 10,000 bootstrap samples. If the *p*-value is less than 10%, we have the evidence that the SGBM fits the data better. If the *p*-value is less than 5%, we have very strong evidence against the commonly-used models.

## 4. Empirical Analysis

### 4.1. Data

The closing prices of the AEX (Netherlands, AEX), BEL 20 (Belgium, BFX), DAX (Germany, GDAXI), CAC 40 (France, FCHI), FTSE 100 (UK, FTSE), Shanghai Shenzhen CSI 300 (China, CSI 300), Nikkei 225 (Japan, N225), SMI (Switzerland, SSMI), and S&P 500 (U.S., SPX) indices are used in the empirical study. The stock markets considered represent nine of the most important stock markets internationally; all of them are mature capital markets (except for Chinese market). As a representative of an emerging market, the Chinese market has been rapidly developing and has had an important influence on the world economy; for these reasons, we also take it into account. The data are taken from the Wind Database. The sample period of eight mature markets is from 1 January 2000

to 31 December 2017, while the starting sample period of CSI 300 is 1 January 2002, as it is the first day that the data are available. First, we use the weekly data over the whole sample period to check the skew phenomenon in stock markets in the long-run. Then, we use daily data in each year to check the skew phenomenon over a year. By rolling forward the data quarterly, we obtain 61 groups of data for Chinese market and 69 groups for the other markets.

Figure 1 and Table 1 illustrate the price series under consideration and show their descriptive statistics, respectively. In Figure 1, we can observe that all stock indices display very similar patterns. For example, during 2004, 2007, and 2016, the indices all clustered at a certain level. It is likely that skew phenomenon exists over these periods.

In Table 1, according to the coefficient of variation (C.V.), we can see that Shanghai Shenzhen CSI 300 index is the most volatile of the indices, while the FTSE 100 is the most stable. Table 2 shows that the nine stock markets are highly correlated, which coincides with the results in Figure 1. The correlation coefficient between the S&P 500 and DAX is the highest, while the correlation coefficient between AEX and CSI 300 is the lowest.

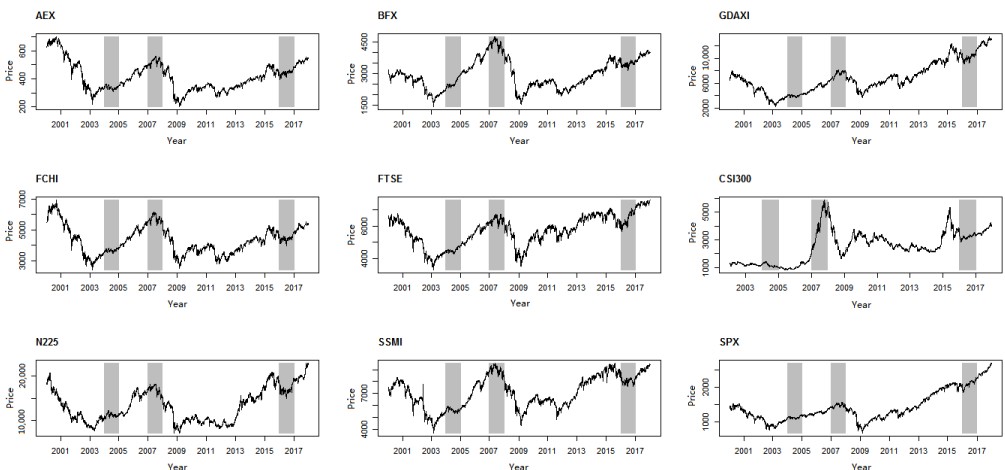

**Figure 1.** Time series of closing price of the nine indices.

**Table 1.** Descriptive statistics of the closing price of the nine indices.

|  | **AEX** | **BFX** | **GDAXI** | **FCHI** | **FTSE** | **CSI 300** | **N225** | **SSMI** | **SPX** |
|---|---|---|---|---|---|---|---|---|---|
| Mean | 416.2 | 2972 | 6915 | 4338 | 5724 | 2518 | 13,379 | 7117 | 1435.7 |
| Median | 398.66 | 2883.43 | 6557.1 | 4277.65 | 5858.7 | 2522.08 | 12,833.64 | 7112.6 | 1315.23 |
| Min | 199.5 | 1527 | 2403 | 2534 | 3492 | 818 | 7173 | 3880 | 683.4 |
| 25% Quntile | 335.34 | 2459.03 | 5003.16 | 3652.45 | 5129.71 | 1341.48 | 10,124.81 | 6026.76 | 1134.52 |
| 75% Quntile | 489.08 | 3522.49 | 8265.18 | 4987.34 | 6429.46 | 3309.46 | 16,479.26 | 8238.56 | 1646.49 |
| Max | 695.2 | 4749 | 13,479 | 6814 | 7688 | 5737 | 22,903 | 9531 | 2683.3 |
| Std.Dev | 104.59 | 701.68 | 2531.23 | 913.02 | 941.39 | 1111.23 | 3760.91 | 1346.95 | 430.06 |
| C.V. | 0.2513 | 0.2361 | 0.3660 | 0.2105 | 0.1645 | 0.4412 | 0.2811 | 0.1893 | 0.2996 |
| Skewness | 0.6163 | 0.3605 | 0.5846 | 0.4326 | −0.3050 | 0.3216 | 0.3891 | −0.0928 | 0.9049 |
| Kurtosis | −0.1010 | −0.6213 | −0.3751 | −0.4855 | −0.6773 | −0.4601 | −0.9990 | −1.0912 | −0.0328 |

**Table 2.** Correlation coefficients of the nine indices.

|  | **AEX** | **BFX** | **GDAXI** | **FCHI** | **FTSE** | **CSI 300** | **N225** | **SSMI** | **SPX** |
|---|---|---|---|---|---|---|---|---|---|
| AEX | 1 | | | | | | | | |
| BFX | 0.6564 | 1 | | | | | | | |
| GDAXI | 0.4377 | 0.6001 | 1 | | | | | | |
| FCHI | 0.9521 | 0.7876 | 0.4744 | 1 | | | | | |
| FTSE | 0.6229 | 0.7174 | 0.9103 | 0.6783 | 1 | | | | |
| CSI 300 | 0.4071 | 0.4408 | 0.6745 | 0.4445 | 0.6085 | 1 | | | |
| N225 | 0.7302 | 0.8307 | 0.7837 | 0.7885 | 0.8033 | 0.4596 | 1 | | |
| SSMI | 0.6703 | 0.8608 | 0.8333 | 0.7670 | 0.9129 | 0.5606 | 0.8885 | 1 | |
| SPX | 0.4389 | 0.6220 | 0.9632 | 0.4565 | 0.8728 | 0.5338 | 0.8173 | 0.8117 | 1 |

*4.2. Empirical Results*

4.2.1. Weekly Data of the Whole Sample Period

We estimate the parameters of SGBM model and test the skew phenomenon using the weekly closing prices of the nine stock indices over the whole sample period. For more details of parameter estimation approach, see Appendix A.

Table 3 gives the results of parameter estimation for the nine indices. We focus on the estimated skew probability $\hat{p}$. For the nine stock indices, the skew probabilities $\hat{p}$ are 0.3343, 0.3669, 0.3754, 0.3204, 0.2748, 0.4715, 0.2984, 0.2712, and 0.3187, respectively. The corresponding skew levels are 329.4386, 2664.4319, 5540.7140, 3921.4128, 4788.8800, 3303.4179, 10,543.1285, 6589.6171, and 1066.8168, respectively. According to the significance test of the skew probability $p$, all these estimates are statistically significantly different from 0.5 at 5% level, other than the $\hat{p}$ of CSI 300, indicating that there are significant skew phenomena in the eight stock markets. According to Table 1, since C.V. is used to measure the dispersion of data, it is not abnormal that the skew phenomenon does not exist in the Chinese market over the whole sample period. Coinciding with the correlation shown in Table 2, all the $\hat{p}$ values are smaller than 0.5, which means that it is more likely for the dynamics of indices to move downward than upward when hitting the skew level $a$. However, the likelihood ratio test gives different conclusions. There are only four markets on which SGBM model strictly significantly outperforms the other three models at the 10% level, while there are four markets on which SGBM does not outperform them. The $p$-values of likelihood ratio tests in Japanese market is 0.100 against GBM and 0.074 against GBM-J, reaching a significant margin. It is hard to say whether SGBM outperforms these two models, since the $p$-values are at such a marginal level. Therefor, we try to find some clues from the graph of the index.

**Table 3.** Empirical results.

| | $\tilde{\mu}$ | $\sigma$ | $a$ | $p$ | $p$-Value (Significance Test of $p$) | $p$-Value (vs. GBM) | $p$-Value (vs. GBM-J) | $p$-Value (vs. GBM-MRS) |
|---|---|---|---|---|---|---|---|---|
| AEX | 0.0484 | 0.2156 | 329.4386 | 0.3343 | 0.009 | 0.037 | 0.071 | 0.040 |
| | (0.0094) | (0.0052) | (18.8989) | (0.0317) | | | | |
| BFX | 0.0542 | 0.1964 | 2664.4319 | 0.3669 | 0.017 | 0.229 | 0.211 | 0.176 |
| | (0.0037) | (0.0046) | (50.5755) | (0.0216) | | | | |
| GDAXI | 0.0809 | 0.2303 | 5540.7140 | 0.3754 | 0.015 | 0.116 | 0.154 | 0.147 |
| | (0.0050) | (0.0054) | (416.2285) | (0.0388) | | | | |
| FCHI | 0.0406 | 0.2118 | 3921.4128 | 0.3204 | 0.009 | 0.222 | 0.170 | 0.203 |
| | (0.0039) | (0.0049) | (145.9391) | (0.0305) | | | | |
| FTSE | 0.0364 | 0.1699 | 4788.8800 | 0.2748 | 0.003 | 0.028 | 0.041 | 0.037 |
| | (0.0022) | (0.0040) | (24.4696) | (0.0292) | | | | |
| CSI 300 | 0.1114 | 0.2643 | 3303.4179 | 0.4715 | 0.277 | 0.212 | 0.200 | 0.197 |
| | (0.0069) | (0.0066) | (1390.6915) | (0.1439) | | | | |
| N225 | 0.0672 | 0.2136 | 10,543.1285 | 0.2984 | 0.004 | 0.100 | 0.074 | 0.012 |
| | (0.0046) | (0.0051) | (92.3691) | (0.0349) | | | | |
| SSMI | 0.0535 | 0.1811 | 6589.6171 | 0.2712 | 0.002 | 0.089 | 0.097 | 0.032 |
| | (0.0043) | (0.0042) | (35.9129) | (0.0314) | | | | |
| SPX | 0.0690 | 0.1714 | 1066.8168 | 0.3187 | 0.001 | 0.016 | 0.071 | 0.053 |
| | (0.0040) | (0.0040) | (17.2439) | (0.0292) | | | | |

Figure 2 shows the historical price data and skew level of Nikkei 225 index, which is 10,543.1285, as shown in Table 3. In Figure 2, we can see that there are two major time intervals during which the skew phenomenon is very significant. The first interval is from 2001 to 2003, during which the process walks through the skew level several times, but the major part is below it. The second interval is from 2009 to 2013. The vertical lines indicate the dates when the extreme price movements occur. It is obvious that extreme price movements occur frequently when the index is near the skew level. As many researchers have devoted considerable effort to correlation between extreme price movement and herding, the herding may be conducive to the explanation of skew phenomenon.

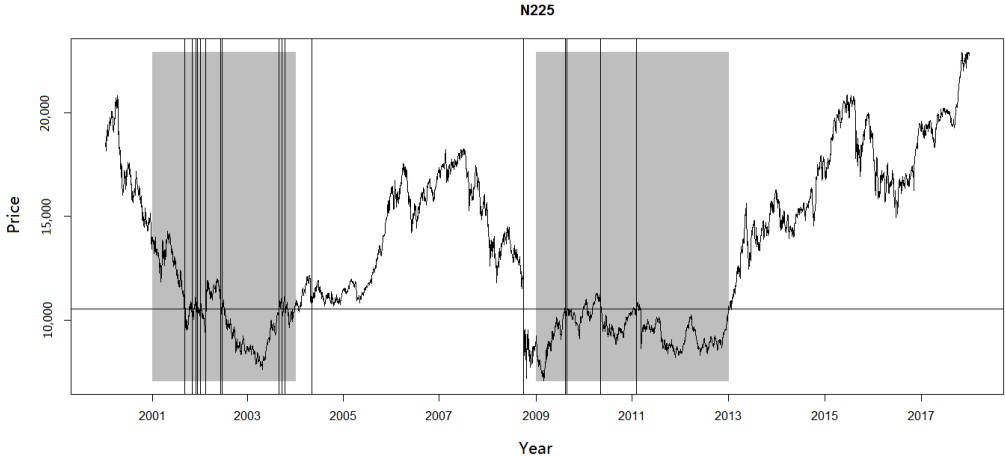

**Figure 2.** Skew level of Nikkei 225 index.

Figure 3 shows the price clustering during 2000–2017, in Nikkei 225 index. As the observations of the index are discrete, it is difficult to find that the dynamic hits a specific level precisely. Thus, we use a small interval around it to stand for the level. Sonnemans [4] was interested in the number of times crossing a level. Besides that, we also consider the number of times reflecting from a level. The difference between "crossing" and "reflecting" is whether the previous price and the following price are on the same side of the level. In Figure 3 we notice that the data exhibit price clustering on some scales. After signalling the skew level in Figure 3 by a straight line, it is obvious that the skew level is one of the points at which the indices cluster. The total number of times hitting the skew level is 53. In addition to three times crossing the skew level from above and three times from below, there are also 17 times reflected by the skew level upward and 30 times downward. For convenience, we assume that, after hitting the skew level, whether the dynamic goes up or goes down can be described by a Bernoulli random variable with probability $p$. The result of hypothesis test shows that the $p$ is rejected to be greater than or equal to 0.5. We can conclude that there is a big chance for Nikkei 225 index to move down when price series hits 10,543.1285, the process goes down in most cases once it touches the skew level.

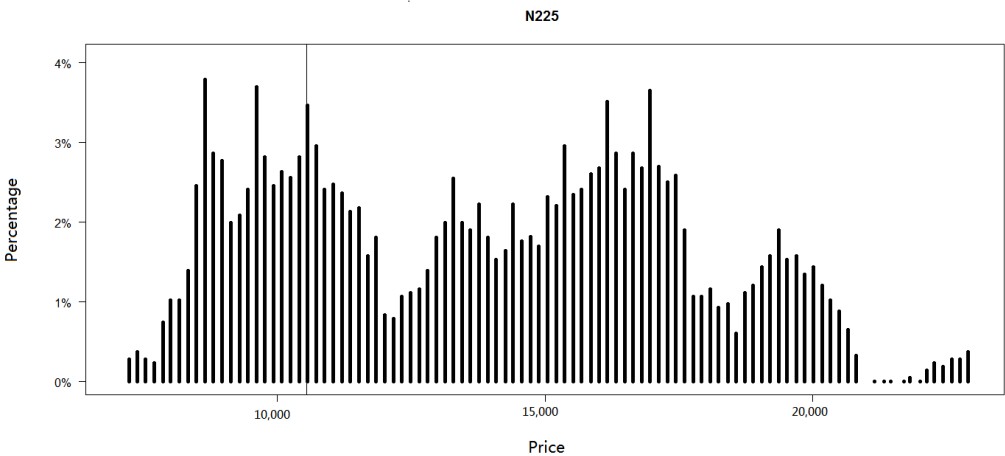

**Figure 3.** Price clustering of Nikkei 225 index.

We can notice that skew level does not need to be a very high level for the skew probability $p$ to be smaller than 0.5, or to be at a very low level for the skew probability $p$ to be bigger than 0.5.

In fact, the skew levels we estimate in the empirical study are all lower than both the means and medians in the eight mature stock markets. However, the skew levels need to be the levels at which the dynamic crosses several times and has a preference for moving upward or downward. Overall, the skew probabilities $p$ are different from 0.5 with statistical significance in the eight markets and the SGBM model does not underperform the standard GBM model, GBM-J model, and GBM-MRS model, proving that skew phenomenon exists broadly in stock markets all around the world.

### 4.2.2. Daily Data over One Year Period

After investigating the skew phenomenon over the whole sample period, we study the skew phenomenon over a one-year period. We obtain 61 subsamples of the CSI 300 index and 69 subsamples of the other eight stock indices. We estimate the parameters and check the skew phenomenon by testing the significance of the skew probability $p$. Table 4 and Figure 4 show the results for the Nikkei 225 index. The detailed results of other indices are available upon request. For simplicity, we only show the results of subsamples whose skew probability $p$ is significantly different from 0.5 at the 10% level. Furthermore, we define the subsamples in which the skew phenomenon exists as skew subsamples.

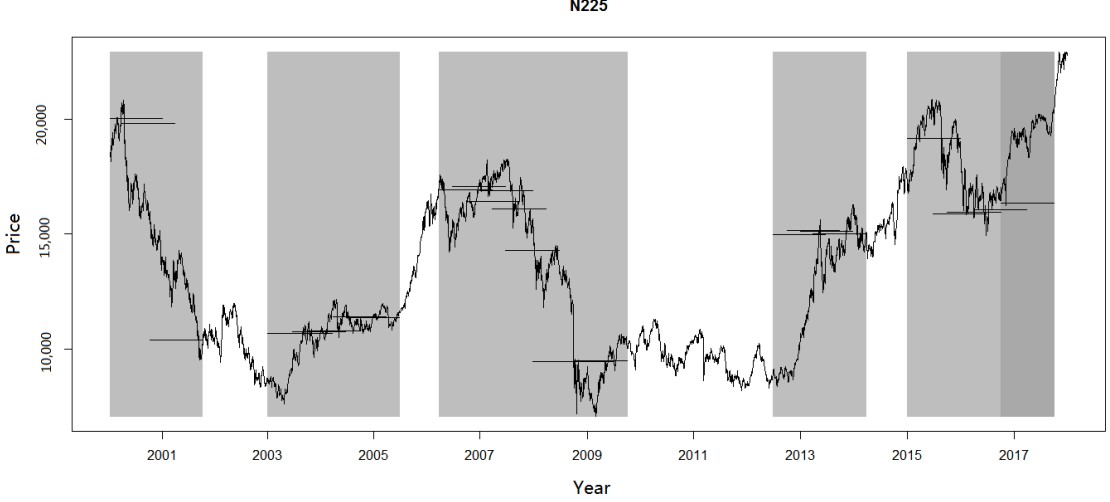

**Figure 4.** Skew levels of the Nikkei 225 index.

There are 28 subsamples in which the skew phenomenon exists, accounting for about 40% of the total subsamples. Almost all the skew probabilities $\hat{p}$ are smaller than 0.5, indicating that these skew levels are resistance levels. The only subsample in which the $\hat{p}$ is bigger than 0.5 is from the fourth quarter of 2016 to the third quarter of 2017, with the corresponding skew level $\hat{a}$ of 16,336.7533. Similar to the weekly data, the skew level of each subsample can be seen as the scale at which the index clusters. Taking the subsample from the second quarter of 2003 to the first quarter of 2004 as an example, Figure 5 shows the conclusion clearly. The skew level we estimate is 10,742.4626. The price clustering phenomenon is also most foremost on the level. Therefore, our skew phenomenon and price clustering phenomenon can confirm with each other.

Table 5 shows the results of the comparison of $p$-values and MSEs for each model. The third column of Table 5 shows that the SGBM model clearly works better than the GBM model in most cases. In detail, the SGBM model wins 19 (67.86%) of the subsamples. The SGBM model also beats the GBM-J model, which is exhibited in the fifth column of Table 5; the SGBM model wins 18 (64.29%) of the subsamples. However, the results indicate that there may not be a noticeable difference between the SGBM model and GBM-MRS model for fitting the data. In fact, the two models capture different characteristics of the dynamic. A combination of them may outperform all the models mentioned here, and this is one of our future works.

**Table 4.** Empirical Results using daily data of the Nikkei 225 index.

|  | $\widetilde{\mu}$ | $\sigma$ | $a$ | $p$ | $p$-Value (Significance Test of $p$) |
|---|---|---|---|---|---|
| 2000 | −0.2456 | 0.2212 | 20,034.8714 | 0.2534 | 0.0030 |
|  | (0.0165) | (0.0101) | (413.4776) | (0.0686) |  |
| 2000Q2-2001Q1 | −0.4097 | 0.2473 | 19,825.2982 | 0.0974 | 0.0039 |
|  | (0.0102) | (0.0113) | (460.0698) | (0.0861) |  |
| 2000Q4-2001Q3 | −0.4189 | 0.2769 | 10,371.7933 | 0.2136 | 0.0554 |
|  | (0.0267) | (0.0128) | (1103.1802) | (0.1588) |  |
| 2003 | 0.3298 | 0.2261 | 10,654.5045 | 0.3531 | 0.0408 |
|  | (0.0406) | (0.0105) | (502.0837) | (0.0811) |  |
| 2003Q2-2004Q1 | 0.5392 | 0.2127 | 10,680.3592 | 0.3660 | 0.0405 |
|  | (0.0488) | (0.0098) | (484.8874) | (0.0801) |  |
| 2003Q3-2004Q2 | 0.4053 | 0.2243 | 10,742.4626 | 0.3860 | 0.0320 |
|  | (0.0530) | (0.0104) | (293.4311) | (0.0691) |  |
| 2003Q4-2004Q3 | 0.1464 | 0.2100 | 10,766.2403 | 0.3933 | 0.0495 |
|  | (0.0470) | (0.0096) | (410.5015) | (0.0580) |  |
| 2004Q2-2005Q1 | 0.0489 | 0.1676 | 11,402.2879 | 0.4020 | 0.0803 |
|  | (0.0268) | (0.0077) | (408.0682) | (0.0746) |  |
| 2004Q3-2005Q2 | 0.0173 | 0.1399 | 11,375.1845 | 0.3919 | 0.0789 |
|  | (0.0252) | (0.0065) | (264.4332) | (0.0785) |  |
| 2006Q2-2007Q1 | 0.0866 | 0.1790 | 16,938.1223 | 0.3330 | 0.0231 |
|  | (0.0357) | (0.0084) | (675.8992) | (0.0694) |  |
| 2006Q3-2007Q2 | 0.2293 | 0.1512 | 17,067.5300 | 0.3369 | 0.0185 |
|  | (0.0235) | (0.0069) | (304.7319) | (0.0620) |  |
| 2006Q4-2007Q3 | 0.1002 | 0.1663 | 16,413.7392 | 0.3096 | 0.0098 |
|  | (0.0367) | (0.0076) | (725.1981) | (0.0720) |  |
| 2007 | -0.0298 | 0.1828 | 16,888.0877 | 0.3503 | 0.0179 |
|  | (0.0389) | (0.0084) | (515.9827) | (0.0608) |  |
| 2007Q2-2008Q1 | −0.1958 | 0.2490 | 16,110.2372 | 0.3452 | 0.0215 |
|  | (0.0230) | (0.01162) | (790.6090) | (0.0691) |  |
| 2007Q3-2008Q2 | −0.1416 | 0.2779 | 14,299.7574 | 0.3758 | 0.0205 |
|  | (0.0618) | (0.0129) | (1274.6560) | (0.0590) |  |
| 2008 | −0.2791 | 0.4525 | 9428.0456 | 0.2776 | 0.0065 |
|  | (0.0472) | (0.0209) | (343.0267) | (0.0798) |  |
| 2008Q2-2009Q1 | −0.2039 | 0.4534 | 9426.7842 | 0.2646 | 0.0039 |
|  | (0.0465) | (0.0212) | (287.9609) | (0.0741) |  |
| 2008Q3-2009Q2 | −0.0461 | 0.4629 | 9426.0125 | 0.2861 | 0.0051 |
|  | (0.0495) | (0.0217) | (325.4943) | (0.0721) |  |
| 2008Q4-2009Q3 | 0.1847 | 0.4524 | 9470.2083 | 0.2678 | 0.0004 |
|  | (0.0471) | (0.0209) | (96.9193) | (0.0522) |  |
| 2012Q3-2013Q2 | 0.5130 | 0.2410 | 14,977.6250 | 0.2404 | 0.0386 |
|  | (0.0201) | (0.0113) | (1109.5259) | (0.1300) |  |
| 2012Q4-2013Q3 | 0.6014 | 0.2586 | 15,166.0798 | 0.2240 | 0.0140 |
|  | (0.0179) | (0.0119) | (660.9910) | (0.1041) |  |
| 2013 | 0.5417 | 0.2663 | 15,113.0267 | 0.3568 | 0.0493 |
|  | (0.0368) | (0.0121) | (1002.0543) | (0.0919) |  |
| 2013Q2-2014Q1 | 0.3197 | 0.2702 | 15,000.5340 | 0.3818 | 0.0676 |
|  | (0.0431) | (0.0124) | (872.2384) | (0.0962) |  |
| 2015 | 0.1548 | 0.2027 | 19,155.8567 | 0.3894 | 0.0726 |
|  | (0.0298) | (0.0094) | (1020.5246) | (0.0744) |  |
| 2015Q3-2016Q2 | −0.1820 | 0.2743 | 15,878.1070 | 0.3056 | 0.0160 |
|  | (0.0334) | (0.0127) | (979.1998) | (0.0894) |  |
| 2015Q4-2016Q3 | 0.0246 | 0.2491 | 15,953.6224 | 0.3582 | 0.0501 |
|  | (0.0331) | (0.0115) | (956.2708) | (0.0814) |  |
| 2016Q2-2017Q1 | 0.2820 | 0.2117 | 16,046.1535 | 0.3235 | 0.0000 |
|  | (0.0238) | (0.0097) | (139.1869) | (0.0362) |  |
| 2016Q4-2017Q3 | 0.1808 | 0.1377 | 16,336.7533 | 0.6801 | 0.0005 |
|  | (0.0071) | (0.0062) | (22.2896) | (0.0370) |  |

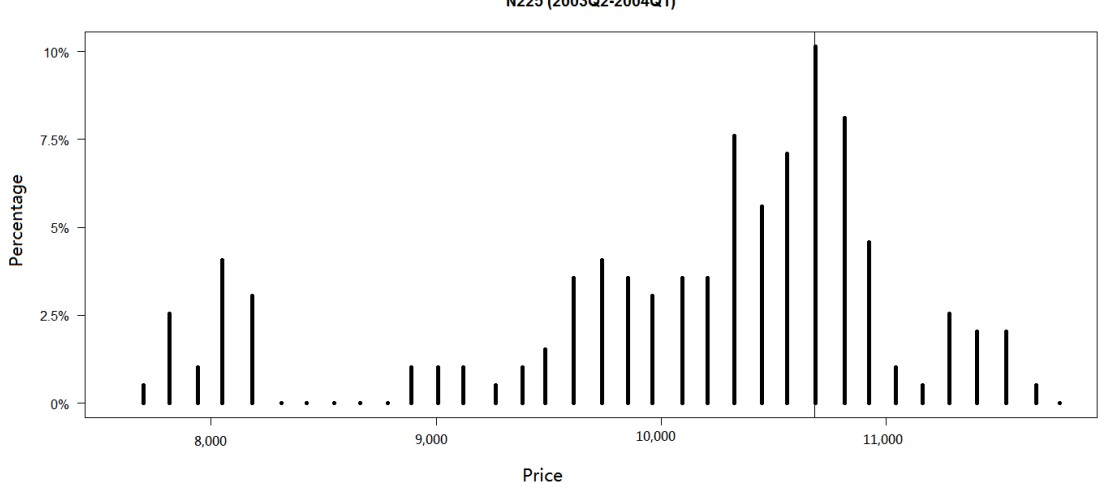

**Figure 5.** Price clustering of Nikkei 225 index (2003Q2–2004Q1).

**Table 5.** Test Results of Nikkei 225 for daily data.

| | GBM | | GBM-J | | GBM-MRS | |
|---|---|---|---|---|---|---|
| | *p*-Value (LR Test) | MSE(SGBM) -MSE(GBM) | *p*-Value (LR Test) | MSE(SGBM) -MSE(GBM-J) | *p*-Value (LR Test) | MSE(SGBM) -MSE(GBM-MRS) |
| 2000 | 0.0124 | −467.1184 | 0.0814 | −59.8494 | 0.0216 | −373.2085 |
| 2000Q2-2001Q1 | 0.0000 | −1007.3783 | 0.0287 | 93.9713 | 0.0000 | −910.8165 |
| 2000Q4-2001Q3 | 0.0113 | −632.1330 | 0.0217 | −359.3064 | 0.0220 | −613.0307 |
| 2003 | 0.0404 | −29.2933 | 0.0806 | −153.4616 | 0.0424 | 469.0911 |
| 2003Q2-2004Q1 | 0.0027 | −80.3167 | 0.0904 | −42.6875 | 0.0719 | −73.0463 |
| 2003Q3-2004Q2 | 0.0879 | 59.5639 | 0.0472 | 7.0149 | 0.0663 | 628.7460 |
| 2003Q4-2004Q3 | 0.0152 | 30.7509 | 0.0178 | −65.8554 | 0.0201 | 389.9553 |
| 2004Q2-2005Q1 | 0.0907 | 6.0259 | 0.0423 | 3.4178 | 0.0715 | 949.0714 |
| 2004Q3-2005Q2 | 0.0728 | −0.5947 | 0.0612 | 6.0508 | 0.0501 | 538.1349 |
| 2006Q2-2007Q1 | 0.0871 | 25.8809 | 0.0695 | −327.3464 | 0.0714 | 329.1633 |
| 2006Q3-2007Q2 | 0.0896 | 56.7752 | 0.0740 | −127.1399 | 0.0733 | 58.6503 |
| 2006Q4-2007Q3 | 0.0318 | 31.5326 | 0.0104 | 38.3551 | 0.0302 | 108.8783 |
| 2007 | 0.0163 | −84.9845 | 0.0261 | 35.8483 | 0.0466 | 31.3096 |
| 2007Q2-2008Q1 | 0.0451 | −207.4157 | 0.0295 | −748.2395 | 0.0691 | −173.9129 |
| 2007Q3-2008Q2 | 0.0320 | −352.3729 | 0.0371 | −218.0522 | 0.0236 | 385.4208 |
| 2008 | 0.0745 | −826.6330 | 0.0185 | −744.8044 | 0.0423 | −543.4644 |
| 2008Q2-2009Q1 | 0.0763 | −319.0402 | 0.0475 | −391.4915 | 0.0563 | −20.1404 |
| 2008Q3-2009Q2 | 0.0701 | −193.5511 | 0.0706 | 359.5804 | 0.0773 | 333.3380 |
| 2008Q4-2009Q3 | 0.0001 | 33.3291 | 0.0311 | −111.1087 | 0.0206 | 174.5915 |
| 2012Q3-2013Q2 | 0.0006 | −212.9670 | 0.0535 | −10.3022 | 0.0233 | −211.3495 |
| 2012Q4-2013Q3 | 0.0003 | −182.9501 | 0.0001 | 95.2869 | 0.0001 | −51.4135 |
| 2013 | 0.0001 | −352.1612 | 0.0513 | 27.2120 | 0.0704 | 1024.1617 |
| 2013Q2-2014Q1 | 0.0067 | 86.0643 | 0.0831 | −220.8618 | 0.0740 | 2345.8511 |
| 2015 | 0.0530 | 26.6559 | 0.0411 | −17.5073 | 0.0519 | 316.1277 |
| 2015Q3-2016Q2 | 0.0103 | −319.1788 | 0.0117 | −72.5074 | 0.0410 | 185.5714 |
| 2015Q4-2016Q3 | 0.0107 | −10.4042 | 0.0192 | 96.3527 | 0.0261 | 3363.1619 |
| 2016Q2-2017Q1 | 0.0078 | −25.4598 | 0.0581 | −15.1377 | 0.0654 | 1218.5149 |
| 2016Q4-2017Q3 | 0.0001 | −287.6788 | 0.0851 | −30.6904 | 0.0752 | −238.8011 |

Table 6 summarizes the results of all the nine indices. It can be easily observed in Table 6 that the SMI has the most skew subsamples among all the indices, followed by the Nikkei 225, while FTSE 100 has the fewest skew subsamples. Since the market behaviors of the different indices vary, it is reasonable that there are more skew subsamples in some markets while fewer in others. Although the numbers of skew subsamples in all markets are fewer than half of total subsamples, the percentages

are still noticeable. Collectively, all the numbers in Table 6 confirm that the skew phenomenon exists in subsamples of daily data over the period of a year. The skew probability $p$ is smaller than 0.5 in most skew subsamples, which means that there are more resistance levels than support levels; this is consistent with the investor psychology: as investors are more sensitive to decline of stock price, stock price will sometimes move downward with a bigger probability when hitting a specific level.

**Table 6.** Empirical results of the nine indices.

|  | AEX | BFX | GDAXI | FCHI | FTSE | CSI 300 | N225 | SSMI | SPX |
|---|---|---|---|---|---|---|---|---|---|
| Total subsamples | 69 | 69 | 69 | 69 | 69 | 61 | 69 | 69 | 69 |
| skew subsamples | 15 | 19 | 14 | 20 | 13 | 23 | 28 | 29 | 20 |
| Percentage(%) | 21.74 | 26.09 | 20.29 | 28.99 | 18.84 | 33.33 | 40.58 | 42.03 | 28.99 |
| $p < 0.5$ subsamples | 15 | 16 | 11 | 17 | 12 | 20 | 27 | 29 | 19 |
| Percentage(%) | 100.00 | 84.21 | 78.57 | 85.00 | 92.31 | 86.96 | 96.43 | 100.00 | 95.00 |
| better than GBM | 10 | 16 | 11 | 14 | 9 | 19 | 19 | 22 | 17 |
| Percentage(%) | 66.67 | 84.21 | 85.71 | 70.00 | 69.23 | 82.61 | 67.86 | 75.86 | 85.00 |
| better than GBM-J | 11 | 11 | 8 | 14 | 7 | 16 | 18 | 15 | 9 |
| Percentage(%) | 73.33 | 57.89 | 57.14 | 70.00 | 53.85 | 69.57 | 64.29 | 51.72 | 45.00 |
| better than GBM-MRS | 7 | 10 | 5 | 12 | 5 | 12 | 9 | 10 | 7 |
| Percentage(%) | 46.67 | 52.63 | 35.71 | 60.00 | 38.46 | 52.17 | 32.14 | 34.48 | 35.00 |

Although there are some markets in which the long-run skew phenomenon does not exist and the percentage of short-run skew subsamples is smaller than 50%, we should not ignore skew phenomenon when modeling asset prices. As a skew model can be reduced to a conventional model, the skew model can be used in both kinds of markets: the markets in which skew phenomenon exists and the markets in which skew phenomenon does not exist. For those markets in which skew phenomenon does not exist, the skew probability will simply be 0.5. However, the conventional model cannot capture skew features, and there will be bias if we use the conventional model to describe markets in which skew phenomenon is present. Thus, it is necessary to introduce the skew model and consider skew phenomenon when modeling asset dynamics.

## 5. Why the Skew Phenomenon Exists

As the market behavior is uncertain and confusing, many researchers try to find some reasons to explain different market phenomena. In this section, we try to explain why the skew phenomenon exists.

As shown in Figures 3 and 5, the skew levels are consistent with the levels that the indices cluster at. Thus, the explanations for price clustering phenomenon can help us to understand the reason for the existence of skew phenomenon. Donaldson and Kim [29] pointed out that the DJIA's rise and fall was indeed restrained by "support" and "resistance" levels; these "support" and "resistance" levels are known as psychological barriers. The existence of such psychological barriers in different markets has been proven in many empirical studies. Sonnemans [4] found that round numbers could act as price barriers for individual stocks. Westerhoff [30] claimed the psychological barriers existed in foreign exchange market. Dowling et al. [31] tested the presence of psychological barriers in WTI and Brent oil futures and found them present in the Brent prices but not in the WTI prices. Skew level is similar to, but not identical to, a psychological barrier. Psychological barriers exist in markets due to investor's perceptions that the fundamental asset value is anchored to a nearest round number. The skew level can be viewed as a psychological barrier, although it is not a round number. Both ideas describe the unusual behavior of asset dynamics when hitting a special level.

According to the empirical results in Tables 1 and 3, the skew probability $\hat{p}$ is smaller than 0.5 and the skew level is lower than the mean in each market. This may be not consistent with what we should expect. Usually, we expect a high skew level $a$ when $p$ is small. However, "running after rising and falling" phenomenon can often be seen in stock markets, which can be connected with herding

behavior and positive feedback trading strategies. Figure 2 illustrates that the skew phenomenon is closely related to extreme price movements. During periods of extreme market movements, the herd behavior is universal in the markets. Devenow and Welch [32] conducted a literature review on the economics of rational herding in financial markets, demonstrating that irrational investors usually disregard their prior beliefs and follow other investors blindly. Chang et al. [33] examined the investor behavior within different international markets (i.e., US, Hong Kong, Japan, South Korea, and Taiwan) and found significant evidence of herding in the two emerging markets (Korea and Taiwan). Perhaps the existence of a low skew level with a small skew probability can be explained by positive feedback trading. When there is a stock market downturn, many investors expect a high probability to keep moving downward. Therefore, a low skew level sometimes correspond to a small skew probability.

Additionally, skew phenomenon may be the result of government regulation. There is less government regulation in stock markets than interest markets or foreign exchange markets, but it does not mean that the government will leave the stock market alone. Stock prices are usually considered to react to external forces. Chen et al. [34] proved that stock returns were exposed to systematic economic news, where many macroeconomic variables would systematically affect stock market returns. To stabilize the stock markets, a government may take some measures and give guidance to markets. Chang et al. [33] proved that, in the emerging market, herd behavior could result due to a relatively high degree of government intervention. In June 2015, the Chinese stock market lost over \$3.2 trillion in value, Chinese government took unprecedented steps to prevent stocks from falling further. Authorities suspended initial public offerings (IPOs), limited bearish bets though CSI 300 Index Futures, and encouraged financial firms to buy more shares. The empirical results show that there was indeed skew phenomenon in 2015. When the stock index hits a specific level, the government will try to guide the trend of market, causing the occurrence of a skew level.

In the end, it is noteworthy that the Chinese market is the only one in which the $p$ is not significantly different from 0.5. As an emerging market, the Chinese stock market started relatively late compared with other developed countries and is still immature. Therefore, it is not abnormal that the long-run skew phenomenon existing in the other mature capital markets can not be found in the Chinese market. Another reason may be that the interest rate in China is relatively high compared with other developed countries. The U.S. held a zero interest rate for seven years, and Europe and Japan have been holding zero or negative interest rate since 2016. However, the interest rate in China was between 4% and 6% during 2000–2018, which was much higher than the interest rates in other areas. Thus, it is inappropriate to ignore time value in the Chinese market. The trend of the dynamic is influenced by the interest rate more significantly in Chinese stock market than in other stock markets. Therefore, the long-run skew phenomenon is also affected by the interest rates, the skew level should be an oblique line rather than a straight line, as the sample period spanned 17 years. When we move to one-year sample period, there is still short-run skew phenomenon in the Chinese market.

## 6. Conclusions

This study tests for the skew phenomenon in nine international stock markets, based on the SGBM model, and find that skew phenomenon is common worldwide. For the weekly data over the whole sample period, the skew probabilities $p$ are significantly different from 0.5 at the 5% level in eight markets. Furthermore, we test the goodness-of-fit of SGBM model and three commonly-used models using the likelihood ratio test. SGBM significantly outperforms other models in four markets: the Dutch market, British market, Swiss market and American market. In the Japanese market, the $p$-value of likelihood ratio test is 0.1, reaching a significant margin. The graph of historical prices show that the Nikkei 225 index goes downward at most times when it hits the skew level. Thus, we can consider the Japanese market to be a skew market as well. Overall, we can say that skew phenomenon exists broadly in the global stock markets.

For the daily data over the one-year period, there are 61 subsamples for the CSI 300 and 69 subsamples for the other indices. There are 15, 19, 14, 20, 13, 23, 27, 29, and 20 skew subsamples in

each market, respectively. The Swiss market has the most skew subsamples while the British market has the fewest skew subsamples. The proportion of skew subsamples, out of the total subsamples, is noticeable. The skew probability is smaller than 0.5 in most skew subsamples, indicating that there are more resistance levels than support levels.

In addition, we attempt to explain why the skew phenomenon exists in stock markets. As the skew levels can be viewed as the barriers at which the indices cluster, psychological barriers of stock price may be one of the reasons. Herding behavior and positive feedback trading strategy may provide another reason, as a skew probability smaller than 0.5 corresponds to a skew level lower than the sample mean in the empirical test. Government regulation can cause the occurrence of skew phenomenon as well.

For the above explanations of the skew phenomenon, an important investment implication is that, besides the characteristics such as jump and regime switching, the skew phenomenon is also noteworthy. In the financial markets with skew phenomenon, the value of skew levels and skew probabilities are great assistance to investors in judging the indices trends. For the government, testing the skew phenomenon is a method to examine whether the intervention is effective. The value of skew levels and skew probabilities are the evidence of the effect of intervention.

**Author Contributions:** All authors have read and agree to the published version of the manuscript. Conceptualization, Y.B. and Z.G.; methodology, Y.B.; software, Y.B.; formal analysis, Y.B.; investigation, Z.G.; data curation, Z.G.; writing—original draft preparation, Z.G.; writing—review and editing, Y.B. and Z.G.

**Funding:** This research was funded by National Natural Science Foundation of China grant number No. 71532001 and No. 11631004.

**Acknowledgments:** The authors are indebted to the participants in the seminar on Stochastic Processes and Financial Engineering at Nankai University for their valuable comments and discussions.

**Conflicts of Interest:** The authors declare no conflict of interest.

## Appendix A. Bayesian Estimation of Skew Brownian Motion Model

We introduce the parameter estimation method in this appendix. If we assume $\Theta$ to be the set of all parameters $(\theta_1, \theta_2, \cdots, \theta_n)$ and the joint prior density function of $\Theta$ to be $p(\Theta)$, the posterior density function of the model is:

$$
\begin{aligned}
p(\Theta|X) &= \frac{l(X|\Theta)p(\Theta)}{\int l(X|\Theta)p(\Theta)d\Theta} \\
&\propto l(X|\Theta)p(\Theta).
\end{aligned}
\tag{A1}
$$

In Bayesian inference, the inference which we want to conduct can be evaluated from the expectation of a certain function $g(\Theta)$:

$$
E\left[g\left(\Theta\right)\right] = \int g\left(\Theta\right)p\left(\Theta|X\right)d\Theta.
\tag{A2}
$$

To avoid the complexity of multiple integrals, the Monte Carlo method is adopted in this paper. Let $\left\{\Theta^{(1)}, \cdots, \Theta^{(m)}\right\}$ be the samples generated from the posterior density function $p\left(\Theta|X\right)$; then, Equation (A2) is approximated by:

$$
E\left[g\left(\Theta\right)\right] = \frac{1}{m}\sum_{i=1}^{m} g\left(\Theta^{(i)}\right).
\tag{A3}
$$

However, due to the difficulty in generating the samples $\left\{\theta^{(1)}, \cdots, \theta^{(m)}\right\}$ directly, we employ the Gibbs sampler, which provides an exercisable way to generate these samples. In fact, the Gibbs sampler always uses the full set of univariate conditionals to define the iteration. In our case, instead of

generating $\Theta^{(i+1)} = \left\{ \theta_1^{(i+1)}, \theta_2^{(i+1)}, \cdots, \theta_n^{(i+1)} \right\}$ from $\Theta^{(i)} = \left\{ \theta_1^{(i)}, \theta_2^{(i)}, \cdots, \theta_n^{(i)} \right\}$ by $p\left(\Theta|X\right)$ directly, we get the $\Theta^{(i+1)}$ with the conditional probability densities as follows:

$$
\begin{aligned}
\theta_1^{(i+1)} &\sim p\left( \theta_1 | X, \theta_2 = \theta_2^{(i)}, \theta_3 = (\theta_3)^{(i)}, \cdots, \theta_n = \theta_n^{(i)} \right), \\
\theta_2^{(i+1)} &\sim p\left( \theta_2 | X, \theta_1 = \theta_1^{(i+1)}, \theta_3 = (\theta_3)^{(i)}, \cdots, \theta_n = \theta_n^{(i)} \right), \\
&\cdots \\
\theta_n^{(i+1)} &\sim p\left( \theta_n | X, \theta_1 = \theta_1^{(i+1)}, \theta_2 = (\theta_2)^{(i+1)}, \cdots, \theta_{n-1} = \theta_{n-1}^{(i+1)} \right),
\end{aligned}
\tag{A4}
$$

To make the Gibbs sampler computationally efficient, the priors are chosen such that the conditional posterior distributions are easy to simulate. According to convention, conjugate priors are used to obtain simple analytical forms. For the resulting posterior distributions, see Chen and Li [35].

For each parameter in SGBM model, the estimation is presented as follows:

*Appendix A.1. Estimation of the Instantaneous Return*

Conditional on $\sigma$, $p$, and $a$, the proper prior distribution of $\mu$ should be a normal distribution $\mathcal{N}\left(\mu_\mu, \sigma_\mu^2\right)$. With the likelihood function in Equation (10), we get the proposal distribution of $\mu$:

$$
\mu | X, \sigma^2, p, a \sim \mathcal{N}\left( \widehat{\mu}_\mu, \widehat{\sigma}_\mu^2 \right),
\tag{A5}
$$

where

$$
\begin{aligned}
\widehat{\mu}_\mu &= \left[ \frac{1}{\sigma^2} \left( \frac{\sum_{i \in N_1} y_{t_i}}{1 - p} + \frac{\sum_{i \in N_2} y_{t_i}}{p} \right) + \frac{\mu_\mu}{\sigma_\mu^2} \right] \widehat{\sigma}_\mu^2, \\
\widehat{\sigma}_\mu^{-2} &= \frac{\Delta t N}{\sigma^2} + \frac{1}{\sigma_\mu^2}.
\end{aligned}
$$

*Appendix A.2. Estimation of the Volatility*

Conditional on $\mu$, $p$ and $a$, the posterior distribution of $\sigma^2$ is $\mathcal{IG}\left(\alpha_\sigma, \lambda_\sigma\right)$. Then, the proposal distribution of $\sigma^2$ is:

$$
\sigma^2 | X, \mu, p, a \sim \mathcal{IG}\left( \widehat{\alpha}_\sigma, \widehat{\lambda}_\sigma \right),
\tag{A6}
$$

where

$$
\begin{aligned}
\widehat{\alpha}_\sigma &= \frac{N}{2} + \alpha_\sigma, \\
\widehat{\lambda}_\sigma &= \frac{\sum_{i \in N_1} \left[ y_{t_i} - (1 - p)\, \mu \Delta t \right]^2}{2 \left(1 - p\right)^2 \Delta t} + \frac{\sum_{i \in N_2} \left( y_{t_i} - p \mu \Delta t \right)^2}{2 p^2 \Delta t} + \lambda_\sigma.
\end{aligned}
$$

*Appendix A.3. Estimation of the Skew Level*

Conditional on $\mu$, $\sigma$, and $p$, we can hardly find the conjugate priors of $a$ to be its prior distribution. Normally, we employ the Griddy–Gibbs sampler according to Ritter and Tanner [36]. Assume that $a$ is uniform on a predetermined interval $(a_l, a_u)$ by observation. Conditional on $\mu$, $\sigma$, and $p$, the density of $a$ is developed:

$$p\left(a|X,\mu,\sigma^2,p\right) \propto \left(\frac{1}{(1-p)\sqrt{2\pi\sigma^2\Delta t}}\right)^{n_1} \exp\left\{-\frac{1}{2(1-p)^2\sigma^2\Delta t}\right.$$

$$\left.\times \sum_{i\in N_1}[y_{t_i}-(1-p)\mu\Delta t]^2\right\} \times \left(\frac{1}{p\sqrt{2\pi\sigma^2\Delta t}}\right)^{n_2} \tag{A7}$$

$$\times \exp\left[-\frac{1}{2p^2\sigma^2\Delta t}\times \sum_{i\in N_2}(y_{t_i}-p\mu\Delta t)^2\right]\frac{1}{a_u-a_l}.$$

We get the cumulative density $F_a$ through the $n$ grid points $\{a_1,\cdots,a_n\}$ and right part of Equation (A7). Then, together with a uniform random number $U_a$ and preselected values $0 < \xi_1 < \cdots < \xi_n < 1$, $a^{(i+1)}$ and the new grid points (i.e., used in the next iteration) are obtained as $F_a^{-1}(U_a)$ and $F_a^{-1}(\xi_i)$.

*Appendix A.4. Estimation of the Skew Probability*

Analogous to the estimation of the skew level, we assume $p$ to be uniform on $[0,1]$ and the grid points to be $\{p_1,\cdots,p_n\}$. For each $p \in \{p_1,\cdots,p_n\}$, conditional on $\mu$, $\sigma$, and $a$, and from the transforms in Equations (5) and (7), we calculate density of $p$ as:

$$p\left(p|X,\mu,\sigma^2,a\right) \propto \left(\frac{1}{(1-p)\sqrt{2\pi\sigma^2\Delta t}}\right)^{n_1} \exp\left\{-\frac{1}{2(1-p)^2\sigma^2\Delta t}\right.$$

$$\left.\times \sum_{i\in N_1}[y_{t_i}-(1-p)\mu\Delta t]^2\right\} \times \left(\frac{1}{p\sqrt{2\pi\sigma^2\Delta t}}\right)^{n_2} \tag{A8}$$

$$\times \exp\left[-\frac{1}{2p^2\sigma^2\Delta t}\times \sum_{i\in N_2}(y_{t_i}-p\mu\Delta t)^2\right].$$

Then, we get the cumulative density $F_p$. The other procedure is the same as what is done to estimate the skew level. Finally, we demonstrate that $p^{(i+1)}$ equals $F_p^{-1}(U_p)$ and the new grid points are $\left\{F_p^{-1}(\xi_1),\cdots,F_p^{-1}(\xi_n)\right\}$.

*Appendix A.5. Simulations*

To evaluate the performance of our SGBM model and estimation approach, we conduct a set of simulations. The simulation results are presented in the following.

We set the parameter values at $\tilde{\mu} = 0.0672$, $\sigma = 0.2136$, $p = 0.2984$, and $a = 10,543.1285$, which are the same as the estimates that we obtain using the price series of Nikkei 225 index, as analyzed in Section 4. For the skew model, the $S_t$ is generated as specified in Equation (1) for the samples of size 1000. As the skew phenomenon is set to exist in the simulated sample, the parameters are expected to be biased if the presence of skew phenomenon is ignored. For the simulated sample, we compare the estimates of SGBM model using the Bayesian approach and the estimates of conventional GBM model using maximum likelihood estimation. the absolute relative error (ARE) of the estimates, between the calculated value and experimental value, is defined as

$$ARE = \left|\frac{\hat{\theta}-\theta}{\theta}\right|, \tag{A9}$$

where $\theta$ is used as a generic notation to denote each parameter in the model. The simulation results are presented in Table A1.

**Table A1.** Simulation Results.

| | $\widetilde{\mu}$ | $\sigma$ | $a$ | $p$ | *p*-Value (Likelihood Ratio Test) | *p*-Value (Significance Test of *p*) |
|---|---|---|---|---|---|---|
| SGBM | 0.0593 | 0.2109 | 10,367.3596 | 0.3034 | 0.04 | 0.0000 |
| | (0.0063) | (0.0049) | (83.2747) | (0.0370) | | |
| ARE | 11.75% | 1.26% | 1.67% | 1.65% | | |
| GBM | 0.0227 | 0.2152 | | | | |
| ARE | 66.22% | 0.75% | | | | |

For the simulated sample, the parameters are estimated to be $\widetilde{\mu} = 0.0593$, $\sigma = 0.2109$, $a = 10,367.3596$, and $p = 0.3034$ under the SGBM assumption, and the parameters are estimated to be $\widetilde{\mu} = 0.0227$ and $\sigma = 0.2152$ under the GBM assumption. We can see that the absolute relative error reached as high as 66.22% between the real value of $\widetilde{\mu}$ and estimated value of $\widetilde{\mu}$ under the GBM model, while the absolute relative error is much smaller under the SGBM assumption. As for $\sigma$, the ARE is small for both models. It turns out to be true that skew phenomenon, if ignored, can yield a substantial bias in the estimates of parameters.

Finally, we consider testing for the presence of skew phenomenon on the basis of the likelihood ratio test and significance test of skew probability $p$. As shown in Table A1, the skew probability is significantly different from 0.5, thus there is skew phenomenon present in the simulated sample. The *p*-value of the likelihood ratio test shows that the SGBM model outperforms the GBM model. We can see that skew phenomenon plays an important role in the dynamics of stock prices, such that it is essential to take into consideration when modeling stock prices.

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
