# Peer review of "An Empirical Investigation to the “Skew” Phenomenon in Stock Index Markets: Evidence from the Nikkei 225 and Others"

_sustainability, doi:10.3390/su11247219_

Round 1
Reviewer 1 Report
I like the model and the attempt to fit it to data.
Relation to literature is not very clear to me after reading the introduction. Exactly how is your work related to Decamps et al.(2004), Decamps et al. (2006a, 2006b)?
The benchmark used to show the superiority of skewed GBM is a poor choice in my view ("We try to find some empirical evidence that skew Brownian motions can fit the market data better than standard Brownian motions.") Standard BM or GBM is no longer used in any serious model. We only uses these models when introducing our students to option pricing. I therefore remain unconvinced that the empirical analysis gives strong results because the null hypothesis is a weak one. Of course the empirical analysis changes completely if the test does not involve only one parameter but structurally different models.
The manuscript would benefit from more stringency. Focus on the main issue! Why are there comments such as "The world economy has been recovering step by step since 2009 in spite of a fluctuation from 2015 to 2016."? This is unrelated to the main issue of the research.
I do not understand the arguments in "Therefore, the skew level should not be a constant but a line with a positive slope because it is an increasing function of time." Skew level is determined ex post, so is a structural parameter. What would happen if you allow an affine linear function rather than a constant level? There is some attempt to shed light on the reason for the existence of a skew level in the last section of the paper. I am not really convinced by these mostly speculative considerations.
I wonder about the impact of looking at log prices. Most models aim to fit return distributions (plus time-dependent volatility and other features). To support your model you mention on page 12: "From Figure 2, we can see that there are much more times that the dynamic goes down than goes up when hitting the skew level." I cannot see this. Maybe a more thorough analysis of returns when prices are below the skew level vs above it might evidence this claim. Of course mean-reversion, given that the skew level is ex post, has to be stripped out to get a statistically meaningful test.
poor proofreading
"posotively", "beceuse" "Quntile" and much more
Reviewer 2 Report
The central idea of the work is very interesting, proposed model are good describe. In my opinion reasons of skew phenomenon should be describe in 'Introduction'. From the editorial point of view paper is good wrote.
Reviewer 3 Report
(1) In the theoretical literature and empirical evidence section, the authors should state clearly what the differences (empirical model, or main contribution, etc.) between this paper and existing literature, and how it compares to the existing literature.
(2) Is this paper suitable for using the skew geometric Brow
nian motions (GBMs) model? The authors should provide the testing results.
(3) Based on the findings of statistical research, more policy implications are needed in the conclusions section.
(4)The writing in this paper is unclear and sometimes confusing, and should be made clearer throughout the paper.
Round 2
Reviewer 1 Report
I find the manuscript much improved. Language needs to be improved further however.
Reviewer 3 Report
The manuscript has been fully revised according to reviewer's opinion. I had no further comment and recommendation.